# Genome-Wide Identification and Expression Patterns of *AcSWEET* Family in Pineapple and *AcSWEET11* Mediated Sugar Accumulation

**DOI:** 10.3390/ijms232213875

**Published:** 2022-11-10

**Authors:** Wenqiu Lin, Yue Pu, Shenghui Liu, Qingsong Wu, Yanli Yao, Yumei Yang, Xiumei Zhang, Weisheng Sun

**Affiliations:** 1South Subtropical Crop Research Institute, Chinese Academy of Tropical Agricultural Sciences, Zhanjiang 524091, China; 2Laboratory of Tropical Fruit Biology, Ministry of Agriculture, Zhanjiang 524091, China; 3Key Laboratory of Hainan Province for Postharvest Physiology and Technology of Tropical Horticultural Products, Academy of Tropical Agricultural Sciences, Zhanjiang 524091, China; 4College of Horticulture, South China Agricultural University, Guangzhou 510642, China

**Keywords:** sugars will eventually be exported transporters (SWEETs), expression patterns, AcSWEET11, sugar accumulation, pineapple

## Abstract

Pineapple (*Ananas comosus* L.) is an important fruit crop in tropical regions, and it requires efficient sugar allocation during fruit development. Sugars Will Eventually be Exported Transporters (SWEETs) are a group of novel sugar transporters which play critical roles in seed and fruit development. However, the function of *AcSWEETs* remains unknown in the sugar accumulation. Herein, 17 *AcSWEETs* were isolated and unevenly located in 11 chromosomes. Analysis of a phylogenetic tree indicated that 17 genes were classified into four clades, and the majority of *AcSWEETs* in each clade shared similar conserved motifs and gene structures. Tissue-specific gene expression showed that expression profiles of *AcSWEETs* displayed differences in different tissues and five *AcSWEETs* were strongly expressed during fruit development. *AcSWEET11* was highly expressed in the stage of mature fruits in ‘Tainong16’ and ‘Comte de paris’, which indicates that *AcSWEET11* was important to fruit development. Subcellular localization analysis showed that AcSWEET11 was located in the cell membrane. Notably, overexpression of *AcSWEET11* could improve sugar accumulation in pineapple callus and transgenic tomato, which suggests that *AcSWEET11* might positively contribute to sugar accumulation in pineapple fruit development. These results may provide insights to enhance sugar accumulation in fruit, thus improving pineapple quality in the future.

## 1. Introduction

Carbohydrates are not only vital to supplying necessary energy during plant growth and development, but they also determine fruit quality and flavour [1]. Carbohydrate in plants are produced in the chloroplast by photosynthesis, and it is then translocated to the sink organs by the long-distance transport system of phloem [2]. The symplasmic pathway, the apoplasmic pathway and the alternation between apoplasmic and symplasmic unloading are potential ways for phloem uploading. However, the apoplasmic phloem uploading needs the assistance of specific sugar transporters. Three types of sugar transporters have been identified to date, including monosaccharide transporters [3], sucrose transporters [4], and Sugars Will Eventually be Exported Transporter (SWEET) proteins [5].

The SWEET family members are the novel proteins of carbohydrate transporters in plants, and they are featured by the MtN3/Saliva (PF03083) motif (known as PQ-loop repeat) with seven alpha-helical transmembrane domains connected by a PQ-loop repeat [6]. SWEETs are comprised of a 3–1–3 TMH structure based on the tandem repeats of two 3-TMH domains by a single transmembrane domain [7]. To date, genome-wide analysis of SWEET genes has identified their presence in plants, including 17 in *Arabidopsis thaliana* [6] (Chen et al., 2010), 21 in rice (*Oryza sativa*) [8], 52 in soybean (*Glycine max*) [9], 33 in apple (*Malus domestica*) [10], 25 in banana (*Musa acuminata* L.) [11], and 17 in grape (*Vitis vinifera*) [12]. SWEETs could be divided into four groups (clades I–IV) [6,8] which have many different functions. Clades I and II are mainly transported hexoses, whilst Clade III comprises transported sucrose [12,13]. Clade IV consists of the vacuolar transporters, which regulate the flux of fructose amongst the tonoplast [14].

SWEETs play an important role in accumulation and transportation of sugar. SWEET could change the sugar composition to improve the sugar accumulation in fruit. In tomato, *SlSWEET1a* is highly expressed in young leaves and regulates glucose accumulation. Glucose contents are reduced after the overexpression of *SlSWEET1a*, and up-regulating the ratio of fructose/glucose in the developing fruit modifies the sugar accumulation pattern [15]. Similarly, the overexpression of *VvSWEET10* could increase the contents of glucose and fructose, which improves the total contents in grape [16]. In pear, *PuSWEET15* increased the content of sucrose after overexpression in fruit, whilst the content of sucrose was decreased after silencing [17]. SWEET uploads carbohydrates from the source to the sink via the transport system of phloem, which regulates sugar accumulation for the fruit. Ko et al. [18] found that SlSWEET15 mediates efflux of sucrose from phloem cells to the fruit apoplasm and then imports it into the storage parenchyma cells during tomato fruit development. Similar results were found in soybean, where GmSWEET15 exports the sucrose from the endosperm to the embryo during seed development [19]. Moreover, AcSWEET2a/2b and AcSWEET16b in carambola (*Averrhoa carambola* L.). [20] and MdSWEET9b and MdSWEET15a in apple [21] could participate in sugar accumulation during fruit development.

Pineapple is an important fruit in tropical regions. Sugar composition and content determine the quality of pineapple fruit. Hao et al. [22] isolated AcSWEETs and then analyzed the expression of fruit development by transcriptome; they speculated that AcSWEETs might play an important role in fruit development. However, the function of AcSWEETs remains largely unknown in sugar accumulation during fruit development. In this study, 17 *AcSWEETs* were isolated, and the expression patterns of these genes in different tissues were analyzed. AcSWEET11 was screened out after we analyzed the expression of these genes during fruit development. The function of AcSWEET11 in sugar content was then demonstrated by the transformation of pineapple callus and tomatoes. Our study will not only contribute to understanding the mechanism of sugar accumulation in fruit crops but also will be helpful in improving the quality of pineapple in the future.

## 2. Results

### 2.1. Identification of AcSWEET Family Genes in Pineapple

A total of 17 and 21 SWEET proteins in *Arabidopsis* and rice, respectively, were obtained from the Unirpot database and were blasted in the pineapple genome (https://genomevolution.org/CoGe/NotebookView.pl?nid=937 (accessed on 3 August 2022)). The conserved domain was validated by NCBI-CDD to verify the AcSWEETs. The results showed that all the AcSWEETs contained two MtN3/saliva or PQ-loop superfamily (Figure 1A). A total of 17 AcSWEETs were identified and unevenly distributed on 11 chromosomes (Table 1, Figure 1B). These genes were named on the basis of *Arabidopsis* homologs. The open reading frame (ORF) lengths of AcSWEETs ranged from 579 bp (AcSWEET2) to 906 bp (AcSWEET12), and their protein length varied from 193 to 302 amino acids. The molecular weights were between 22.29 kDa (AcSWEET2) and 33.36 kDa (AcSWEET12). The theoretical isoelectric point (pI) ranged from 6.53 (AcSWEET12) to 9.75 (AcSWEET5), and most of the pI were more than 8.5 except for AcSWEET2, AcSWEET12 and AcSWEET14. The transmembrane regions of AcSWEET were predicted by TMHMM software. The results showed that 13 members of the encoded protein contained seven transmembranes, and two members harbored six transmembranes, whilst two members only had three transmembranes (Appendix A). Further analysis of putative subcellular localization showed that most of the genes were located in the cell membrane, except AcSWEET5 and AcSWEET13, which were located in the nucleus and chloroplast, respectively. Moreover, AcSWEET15 was located not only in the cell membrane but also in the golgi apparatus (Table 1). These results indicated that AcSWEETs may have a membrane protein function.

### 2.2. Phylogenetic Tree Analysis of SWEET in Pineapple, Rice and Arabidopsis

A phylogenetic tree of 17, 17 and 21 SWEETs was constructed using MEGA6 software to study the evolutionary relationships of SWEET proteins amongst pineapple, *Arabidopsis* and rice, respectively. The phylogenetic tree showed that these genes were classified into four clades (I–IV) (Figure 2 and Figure 3A). Clades II and III in *Arabidopsis* and rice contained a large number of SWEET family members, with five and six, respectively. In the meantime, clades I and IV had three members. Most SWEETs were grouped into the same branches between rice and pineapple, which implied that the evolutionary relationships of rice and pineapple were close.

### 2.3. Analysis of Gene Structural and Conserved Motifs in AcSWEETs

The analysis of intron–exon structures showed that the numbers of intron–exon were highly conserved in *AcSWEETs*. All *AcSWEETs* contained several exons, which varied from five to seven. A total of 12 members of *AcSWEETs* harbored six exons. *AcSWEET2*, *AcSWEET3* and *AcSWEET16* contained five exons. *AcSWEET3* and *AcSWEET17* displayed seven exons. Interestingly, AcSWEET17 had a special structure, with six axons composed of two long introns and four short introns (Figure 3C). A total of 10 motifs were identified in *AcSWEETs* using MEME. Three to nine motifs were detected in *AcSWEETs*. *AcSWEET2* had the shortest and least amino acid sequence with three motifs. By contrast, *AcSWEET13* had the longest sequence, with nine motifs. Motif1 and motif3 were observed in all *AcSWEETs*. Motif2, motif4 and motif5 were present in all *AcSWEETs* except *AcSWEET2* and *AcSWEET13*. Motif10 was the specific motif in *AcSWEET9*, *AcSWEET10*, *AcSWEET12* and *AcSWEET13*, which were classified in the same clade (Figure 3B).

### 2.4. Expression Profiles of AcSWEETs in Different Pineapple Tissues 

The expression levels of *AcSWEETs* were analyzed in root, stem, leaf, flower, carpopodium, mature fruit and carpodermis by qRT-PCR to understand the potential functions of these genes (Figure 4). These results indicated that *AcSWEET1*, *AcSWEET10* and *AcSWEET13* were highly expressed in root. *AcSWEET4* and *AcSWEET16* were highly expressed not only in root but also in carpopodium. The expression levels of *AcSWEET2* and *AcSWEET15* were higher in flower than in other tissues. *AcSWEET3*, *AcSWEET7*, *AcSWEET11*, *AcSWEET12* and *AcSWEET14* were strongly expressed in fruit. AcSWEET5 was predominantly expressed in the carpodermis. Interestingly, all *AcSWEETs* had lower expression in leaf and stem. These results indicated that *AcSWEETs* may be involved in the unloading of pineapple.

### 2.5. Expression Patterns of AcSWEETs during Pineapple Fruit Development

*AcSWEETs* significantly affected plant development by regulating carbohydrate transport. *AcSWEETs* were expressed in fruits during development at 0, 7, 21, 35, 49, 63 and 77 days after anthesis. *AcSWEET9* and *AcSWEET13* exhibited high expression during fruit development, whilst *AcSWEET1*, *AcSWEET2*, *AcSWEET4* and *AcSWEET15* exhibited low expression. *AcSWEET3*, *AcSWEET5*, *AcSWEET7*, *AcSWEET10*, *AcSWEET12* and *AcSWEET16* exhibited high expression at 0 and 21 days, but they showed low expression during pineapple fruit development. In the meantime, *AcSWEET11* and *AcSWEET14* had the lowest expression at 0 days, and then their expression increased during fruit development (Figure 5). Further analysis of two pineapple varieties revealed that *AcSWEET11* showed high expression in the stage of mature fruits in ‘Tainong16’ and ‘Comte de paris’ (Appendix A). These results indicated that *AcSWEET11* might play a critical role in pineapple fruit development.

### 2.6. Subcellular Localization of AcSWEET11 Protein

The coding sequence of *AcSWEET11* was fused with *EGFP* driven by 35S promoter to explore the subcellular localization of the AcSWEET11, and the gene was transiently expressed in the cells of *Nicotiana benthamiana*. The transient expression of AcSWEET11 showed that the GFP fluorescence was specifically observed in the cell membrane (Figure 6). This result indicated that the function of AcSWEET11 was predominantly at the membrane of plant cells.

### 2.7. Overexpression of AcSWEET11 Enhances the Accumulation of Soluble Sugar 

The p*CAMBIA1300-AcSWEET11*-EGFP vector was transformed into an embryogenic callus of pineapple by *agrobacterium* -mediated vacuum infiltration to further confirm the function of AcSWEET11. The callus was used to detect the soluble sugar after co-cultivation for 3 days. The result showed that transient callus (18.01 mg/g FW) had 1.4 times the content of soluble sugar as a normal callus (12.81 mg/g FW). The expression of transient calluses were significantly higher than normal calluses (Appendix A). Therefore, AcSWEET11 plays an essential role in sugar accumulation of pineapple.

The p*CAMBIA1300-AcSWEET11*-EGFP constructor was transferred into micro-tom tomatoes to gain further insight into the function of *AcSWEET11*. Independent transgenic lines (15-1) were obtained from hygromycin B resistance selection with genomic PCR detection. The result showed that soluble sugar contents had significant differences between transgenic lines (15-1) and WT in green and mature fruits. The content of green fruit was 9.07 mg/g FW in WT, whilst it was 16.51 mg/g FW in 15-1, which was 1.8 times as many as that of WT. In mature fruits, the content of 15-1 was 20.33 mg/g FW, which was significantly higher than that of WT (11.63 mg/g FW). Intriguingly, green fruit of 15-1 accumulated significantly more soluble sugar than WT mature fruits (Figure 7). These results suggest that *AcSWEET11* enhanced soluble sugar accumulation in fruit.

## 3. Discussion

Carbohydrates play an important role in plant growth and development [16,18], and their allocation and accumulation determine the fruit quality during development [1]. SWEET genes are a novel class of sugar transporters that move sugars between tissues by the phloem [23]. Hitherto, genome-wide analysis showed that SWEETs were separated in many plants, such as rice [8], apples [10], banana [11], tomato [15], cucumbers [20] and longan [24]. In our study, 17 SWEETs were identified from the pineapple genome, which is different from a previous report that isolated 18 in MD-2. This difference is likely due to the different criteria used in identifying the candidate genes. Hao et al. [22] believed that if only the genes have MtN3/saliva and PQ-loop domains, they will belong to the SWEET family. In the present study, the candidate members contained either of two MtN3/saliva or the PQ-loop structural domain, and the lengths of amino acids were from 100 aa to 400 aa from the MD-2 genome. Aco006158.1 contained MtN3/saliva and PQ-loop domains in addition to the PQQ-DH-like superfamily, and the length of amino acids was 885 aa, which was larger than that of SWEET members. Therefore, Aco006158.1 did not belong to the SWEET family. Furthermore, Hao et al. [22] found 21 members in F153, which shows that the difference in the numbers of genes is due to the genome of different varieties.

SWEET proteins play key roles in plant growth. Chen et al. [25] and Guo et al. [26] have found that *AtSWEET2* and *AtSWEET17* were highly expressed in root, but they perform a different function. Further study showed that *AtSWEET2* limits carbon sequestration in root, whilst *AtSWEET17* was a fructose uniporter in the root of *Arabidopsis*. Similarly, the interaction between *StSWEET11* and StSP6A triggers the efflux of sucrose to regulate tuber formation of potato [27]. In our study, *AcSWEET1*, *AcSWEET4*, *AcSWEET10*, *AcSWEET13* and *AcSWEET16* were highly expressed in root (Figure 4), suggesting that these genes may have effected on sugar transported in pineapple root. SWEET not only affects root formation but also plays an important role in floral organ development and flowering time. In *Jasminum sambac*, *SWEET5* is upregulated during flower opening and sexual reproduction [28]. In turn, the overexpression of *AtSWEET10* can significantly accelerate flowering in *Arabidopsis* [29]. OsSWEET11 affects pollen development [30], whilst *AtSWEET15* is expressed during pollen maturation and germination [31]. In our study, *AcSWEET2* and *AcSWEE15* were more highly expressed in flower compared to other tissues, which indicates their role in flower formation or florescence (Figure 4). Moreover, *AcSWEET3*, *AcSWEET7*, *AcSWEET11* and *AcSWEET14* were strongly expressed in fruit (Figure 4). Similar results were found in pears [17], loquats [32] and grapes [16], which imply their key role in pineapple fruit. Further analysis of expression in pineapple fruit development demonstrated that AcSWEET3 and AcSWEET7 were highly expressed at 0 days, and they then decreased with fruit development. In the meantime, the expression levels of *AcSWEET11* and *AcSWEET14* increased. The expression in two varieties exhibited that AcSWEET11 played a central role during pineapple fruit development (Appendix A). Interestingly, SWEET is involved in phloem loading from the source leaves [6]. However, in our study, *AcSWEETs* had low expression in leaf and stem. These results suggest that AcSWEETs transported sugars, and might work together with other sugar transporters in pineapple, such as sucrose carriers. Moreover, *AcSWEET6*, *AcSWEET8*, *AcSWEET9* and *AcSWEET17* have not yet been detected in specific tissues including root, leaf, stem, flower, carpopodium, carpodermis, and mature fruit; these results indicate that *AcSWEET6*, *AcSWEET8*, *AcSWEET9* and *AcSWEET17* might be expressed in other tissues of pineapple. 

The ability of transmembrane transport would determine sugar accumulation in fruit. SWEET is one of the most important sugar transporters, and it can transport sucrose, glucose and fructose [5,33,34]. Several reports have shown that SWEET improved sugar contents in different plants, such as apple [10], banana [11] and grape [16]. Recent studies have found that SWEET mediated sugar efflux across the cell membrane following the concentration gradient without supplementary energy [35]. SlSWEET15 regulates sucrose efflux from phloem cells to the fruit apoplasm, and then it imports it into the storage parenchyma cells during fruit development in tomato [18]. In grape, the levels of glucose, fructose and total sugar increased significantly after the overexpression of VvSWEET10 in grapevine calli and tomatoes, which implied that VvSWEET10 could improve sugar accumulation [16]. Further study demonstrated that sucrose/H+ exchangers may efflux cytosolic sugars in fruit to alleviate the demand for supplementary energy of sugar transport at the plasma membrane [36]. These results suggest that SWEETs regulated sugar efflux from cell membranes to control sugar accumulation in fruit development. Herein, AcSWEET11 was located in the cell membrane, whilst sugar contents were enhanced after the overexpression of AcSWEET11 in pineapple callus and tomatoes. These results indicate that AcSWEET11 may regulate sugar transportation and subsequently lead to sugar accumulation in fruit.

## 4. Materials and Methods

### 4.1. Plant Materials

‘Comte de paris’ and ‘Tainong16’ were cultured in the South Subtropical Crop Research Institute of the Chinese Academy of Tropical Agricultural Sciences (21_1002” N; 110_16034” E). Different tissues were collected including root, leaf, stem, flower, carpopodium, carpodermis, young fruit, immature fruit and mature fruit, and they were stored at −80 °C. The induction of embryogenic callus were conducted following Lin et al. [37].

### 4.2. Genome-Wide Identification of SWEET Family in Pineapple

The protein sequences of SWEET in *Arabidopsis* and rice were downloaded from the Unirpot database (https://www.uniprot.org/uniprotkb?facets=model_organism%3A3702%2Creviewed%3Atrue&query=sweet (accessed on 3 August 2022)). Related information of SWEETs is shown in Appendix A. The candidate protein sequences were identified from pineapple genome based on a homolog of *Arabidopsis* SWEET sequences by TBtools [38]. Thereafter, the conserved domain database (NCBI-CDD, https://www.ncbi.nlm.nih.gov/cdd (accessed on 11 August 2022)) was used to confirm whether the candidate proteins have one or more MtN3/saliva. Subsequently, these sequences were scanned by TBtools to visualise the common conservative motif. Ultimately, 17 AcSWEETs were identified (Table 1). These *AcSWEETs* in pineapple were named according to their phylogenetic relationships to the members of *Arabidopsis*. The molecular weight (Mw) and theoretical (pI) of AcSWEET proteins were predicted by the Expasy tool (https://www.expasy.org/ (accessed on 20 August 2022)). Subcellular localization was predicted by Plant-mPLoc (http://www.csbio.sjtu.edu.cn/cgi-bin/PlantmPLoc.cgi (accessed on 21 August 2022)). Chromosomal localization was drawn by TBtools [38] based on the starting and stop positions of all SWEETs in pineapple. The transmembrane helices of AcSWEET protein sequences were predicted by TMHMM (http://www.cbs.dtu.dk/services/TMHMM/ (accessed on 24 August 2022)).

### 4.3. Analysis of Conserved Motif and Gene Structure

The conserved motifs of 17 AcSWEETs were analysed using the MEME software version 5.3.2 (https://meme-suite.org/meme/meme_5.3.2/tools/meme (accessed on 25 August 2022)). Gene structures were analyzed according to the exon/intron of *AcSWEETs*. The schematics of conserved motifs and gene structures were visualized using TBtools software.

### 4.4. Sequences Alignment and Phylogenetic Tree Analysis

The amino sequences of SWEET genes in pineapple, *Arabidopsis* and rice were obtained for multiple alignments using the MUSCLE alignment module in MEGA6 with default parameters. The data matrix of alignment was used to generate a phylogenetic tree with the value of 1000 bp replicates by the neighbor-joining method [39]. 

### 4.5. Analysis of Gene Expression

Total RNA extraction and qRT-PCR were conducted following Lin [40]. qRT-PCR was conducted on a LightCycler^®^ 480 II (Roche, Basel, Switzerland) using SYBR Green qPCR Master Mix (Thermo Fisher Scientific, Waltham, MA, USA). Specific primers were designed using Primer-BLAST programme (https://www.ncbi.nlm.nih.gov/tools/primer- blast/ (accessed on 13 February 2019)) of NCBI. The primer sequences were listed in Appendix A. Relative gene expression was calculated using the 2^−ΔΔCT^ method with the pineapple β-actin gene as the internal reference gene. All analyses were performed in three biological replicates.

### 4.6. Subcellular Localization of AcSWEET11

The CDS of *AcSWEET11* was conducted into p*CAMBIA1300*-35S-EGFP without a stop condone by *BamH* I and *Sal* I digestion. Subsequently, the plasmid p*CAMBIA1300-AcSWEET11*-EGFP was transferred into *A. tumefaciens* strain EHA105 and was injected into tobacco leaves. After 2 days, the GFP fluorescence of these tobacco leaves was observed by confocal laser scanning microscopy. The primers of p*CAMBIA1300*-35S-EGFP vector are provided in Appendix A.

### 4.7. AcSWEET11 Overexpression in Pineapple Embryogenic Callus and Tomato

The plasmid p*CAMBIA1300-AcSWEET11*-EGFP underwent transient transformation into embryogenic callus of pineapple by *agrobacterium* vacuum infiltration following Luan [41]. After 3 days of co-culture, the sugar contents of these callus were detected.

Plasmid p*CAMBIA1300-AcSWEET11*-EGFP was transformed into tomatoes by *agrobacterium* strain EHA105 to probe the function of AcSWEET11 in fruit. The plants were generated following Sun et al. [42]. The transgenic tomato plants were identified by detection with the primers of hygromycin (Appendix A). Transgenic tomato plant T1 lines named as 15-1 were obtained for further analysis.

### 4.8. Analysis of Sugar Content

The total soluble sugar of tomato fruits and pineapple callus were analyzed using the anthrone-sulphuric acid colorimetry method [43]. Data are presented as the mean of ± SDs. Significant analysis was carried out using SPSS software (version 16.0) and one-way ANOVA test with a Tukey’s test (*p* < 0.05).

## 5. Conclusions

A total of 17 *AcSWEETs* were identified in the pineapple genome, and they were clustered into four clades. Tissue-specific expression showed that AcSWEET genes may play different roles in the development of pineapple tissues and that AcSWEET11 might be involved in fruit development. Furthermore, the overexpression of AcSWEET11 has improved sugar accumulation in pineapple callus and transgenic tomato. This work will contribute to understanding the functional characteristics of AcSWEETs and improving the quality of pineapple.

## Figures and Tables

**Figure 1 ijms-23-13875-f001:**
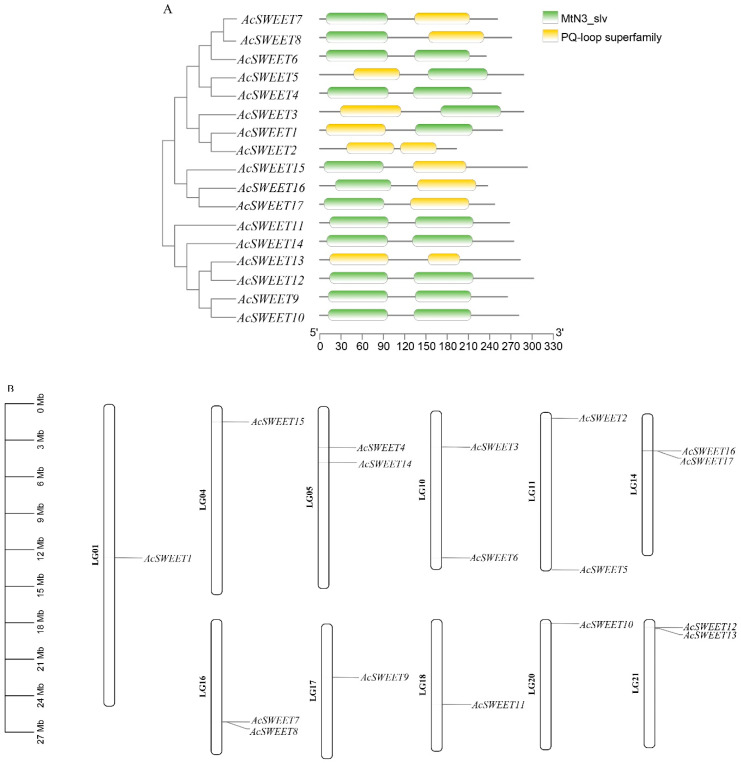
Conserved domain analysis of AcSWEETs and chromosomal location. (**A**) Different domains are represented in different colors along with the conservative domain of prediction as shown in the legend. (**B**) Location of AcC5-MTase genes on the chromosomes.

**Figure 2 ijms-23-13875-f002:**
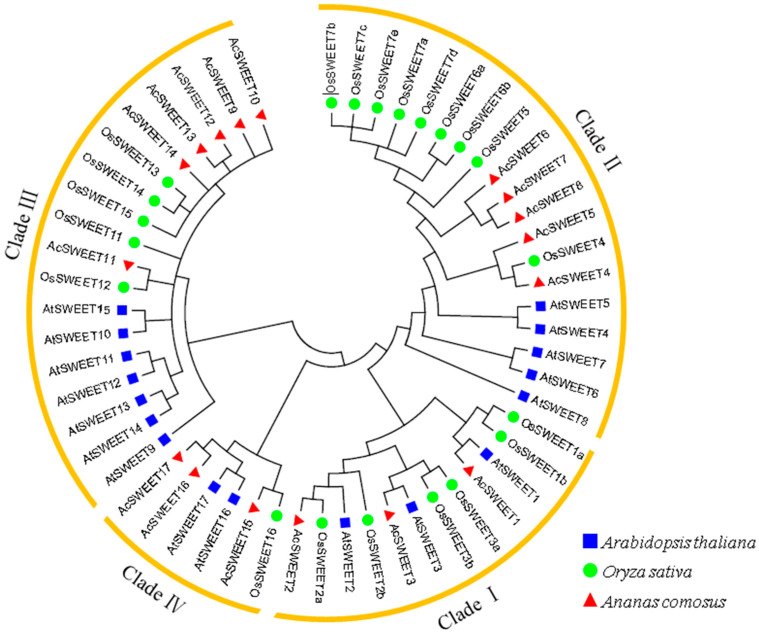
Phylogenetic tree of SWEETs from *Arabidopsis*, rice and pineapple. We used the neighbor-joining method to draw phylogenetic the tree with MEGA6.0 with 1000 bootstraps replicates. Four subgroups were divided as Clades I–IV.

**Figure 3 ijms-23-13875-f003:**
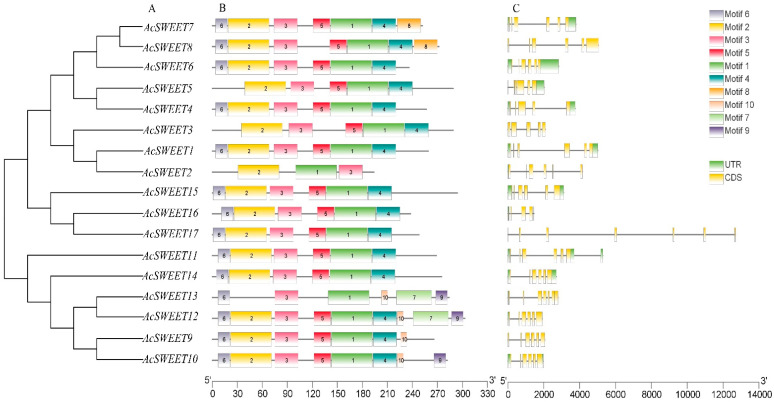
Evolutionary, gene structural and motif analyses of *AcSWEETs* in pineapple. (**A**) phylogenetic tree of candidate amino acid sequences was constructed using the neighbor-joining method with 1,000 bootstraps in MEGA 6.0. (**B**) The conserved motifs of AcSWEETs proteins were analyzed using the MEME tool. (**C**) The gene structure of *AcSWEETs* was identified by TBtools.

**Figure 4 ijms-23-13875-f004:**
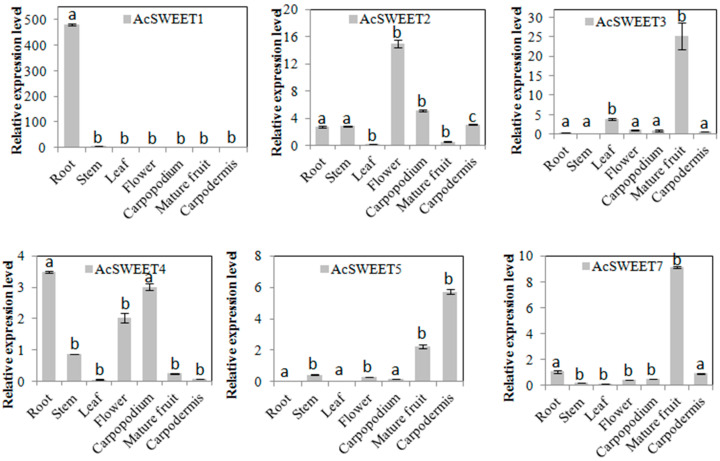
Tissue-specific expression profiles of *AcSWEETs* in pineapple. Values represented as mean of three biological replicates, and standard deviation bars represented as means ± SD (n = 3). Different letters indicate significant differences at *p* < 0.05.

**Figure 5 ijms-23-13875-f005:**
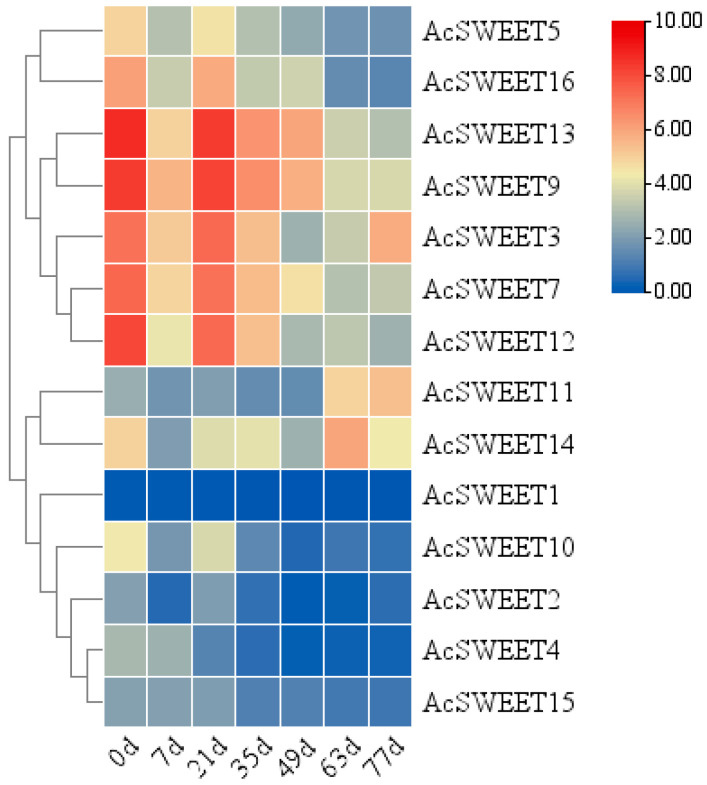
Expression analysis of *AcSWEETs* during ‘Comte de paris’ fruit development conducted by qRT-PCR. d, days after anthesis.

**Figure 6 ijms-23-13875-f006:**
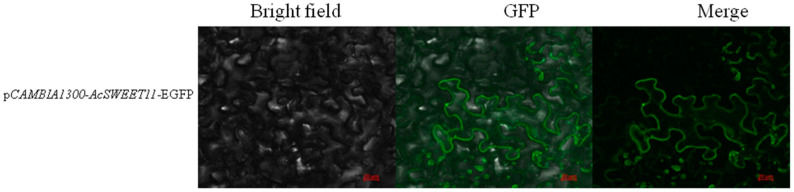
Subcellular localization analysis of AcSWEET11 in *Nicotiana benthaminana* leaves after 2 days of *agrobacterium* injection. Scale bars = 20 μm.

**Figure 7 ijms-23-13875-f007:**
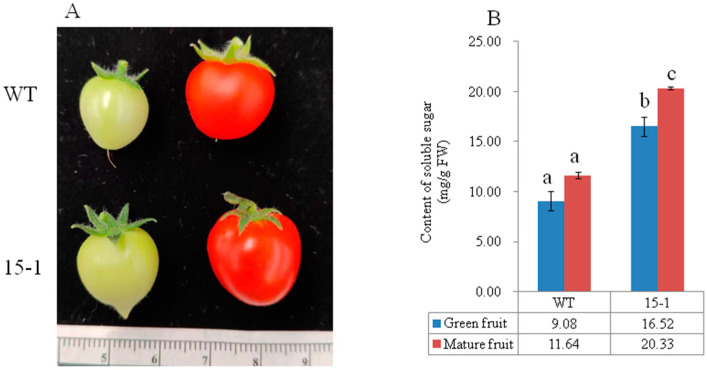
Overexpression of *AcSWEET11* improved soluble sugar in micro-tom tomato. (**A**) Development of tomato fruit in WT and transgenic tomato lines (15-1). (**B**) Soluble sugar content of WT and transgenic tomato lines (15-1). Blue columns presented in green fruit, red columns presented in mature fruit. Values denoted three biological replicates, and vertical lines represented as mean of SD. Different letters denoted significant difference with one-way ANOVA test by Tukey’s test (*p* < 0.05).

**Table 1 ijms-23-13875-t001:** Overview of AcSWEET genes identified in pineapple.

Gene ID	Gene Name	Chromosome Location	ORF Length (bp)	Exon No.	Amino Acid (aa)	Molecular Weight (kDa)	Theoretical pI	Predicted Subcellular Localization
Aco011302.1	*AcSWEET1*	LG01:12,586,879 –12,591,888	774	6	258	28.90	9.21	Cell membrane
Aco016508.1	*AcSWEET2*	LG11:483,467 –487,615	579	6	193	22.29	7.67	Cell membrane
Aco010708.1	*AcSWEET3*	LG10:2,953,844 –2,955,929	864	5	288	32.68	9.34	Cell membrane
Aco004463.1	*AcSWEET4*	LG05:3,365,524 –3,369,279	768	5	256	28.14	9.16	Cell membrane
Aco005793.1	*AcSWEET5*	LG11:12,923,783 –12,925,798	864	6	288	32.61	9.75	Nucleus
Aco016418.1	*AcSWEET6*	LG10:12,052,360 –12,055,189	705	6	235	25.98	8.5	Cell membrane
Aco006156.1	*AcSWEET7*	LG16:8,409,892 –8,413,689	753	6	251	27.39	9.2	Cell membrane
Aco006155.1	*AcSWEET8*	LG16:8,414,666 –8,419,723	813	6	271	29.61	9.18	Cell membrane
Aco003627.1	*AcSWEET9*	LG17:4,381,873 –4,383,930	795	6	265	29.65	9.01	Cell membrane
Aco019048.1	*AcSWEET10*	LG20:312,384 –314,354	843	6	281	31.57	8.93	Cell membrane
Aco001900.1	*AcSWEET11*	LG18:6,983,156 –6,988,444	804	6	268	29.77	9.01	Cell membrane
Aco017831.1	*AcSWEET12*	LG21:646,165 –648,081	906	6	302	33.36	6.53	Cell membrane
Aco016039.1	*AcSWEET13*	LG21:737,444 –740,233	849	7	283	31.73	9.19	Chloroplast
Aco004628.1	*AcSWEET14*	LG05:4,577,567 –4,580,270	822	6	274	30.92	7.66	Cell membrane
Aco002476.1	*AcSWEET15*	LG04:1,314,296 –1,317,407	879	6	293	32.11	9.5	Cell membrane and Golgi apparatus
Aco006347.1	*AcSWEET16*	LG14:3,034,035 –3,035,486	711	5	237	25.75	9.23	Cell membrane
Aco006346.1	*AcSWEET17*	LG14:3,039,676 –3,052,361	741	7	247	26.84	8.78	Cell membrane

## Data Availability

Not applicable.

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
