# Peer review of "Genome-Wide Identification and Expression Patterns of AcSWEET Family in Pineapple and AcSWEET11 Mediated Sugar Accumulation"

_ijms, 2022, doi:10.3390/ijms232213875_

Round 1

Reviewer 1 Report

Dear Editor,
I am pleased to tell you that I have finished the review process of the manuscript
with full title: “Genome-wide identification and expression patterns of AcSWEET family in pineapple and AcSWEET11 mediated sugar accumulation” (ID: ijms-1969756). The manuscript is suitable for publication in journal International Journal of Molecular Sciences.

In this paper is described the importance of carbohydrates for the growth and development of plants, with emphasis of their influence on the fruit quality and flavour. It is stated that they exist three types of sugar transporters: monosaccharide transporters,  sucrose transporters and Sugars Will Eventually be Exported Transporter (SWEET) proteins. The ability of transmembrane transport is very important because it affects the accumulation of sugar in the fruit. The introduction provides an overview of the literature in accordance with the subject of the research. This chapter is well written and provides information on the expression of SWEETs in different plants during fruits and seed development. The Materials and methods sections describe the method for protein sequence identification in pineapple. Methods for extraction of total RNA and qRT-PCR are described, as well as for identifying sequences in the pineapple genome, conserved motifs of AcSVEET, structure and gene expression. Sequence alignment and comparative phylogenetic tree analysis are explained. The results is divided into seven sub-chapters and in them a detailed presentation of the results obtained during the research is given. In the results section, tables and figures are attached on the overview of AcSWEET genes identified in pineapple, conserved domain analysis of AcSWEETs and chromosomal location, phylogenetic tree of SWEETs from arabidopsis, rice and pinapple, evolutionary, gene structural and motifs analyses of AcSWEET genes in pineapple, about tissue-specific expression profiles of AcSWEETs in pineapple and expression analysis of AcSWEETs during fruit development. It was also established subcellular localization analysis of AcSWEET11 in Nicotiana benthaminana leaves after Agrobacterium injection and overexpression of AcSWEET11 in microtom tomato improved soluble sugar. In the discussion, the interpretation of the obtained experimental results was presented, as well as their comparison with existing data from the literature. In accordance with the set goal, they are in conclusions summarized results obtained within this research. The article refers to the adequate literature from leading world journal.

I have decided that the paper can be accepted for publication in journal International Journal of Molecular Sciences after a minor revision.

Thank you for the invitation to participate in the manuscript review process in the journal International Journal of Molecular Sciences.

Kind regards,
S. Milošević

I’m giving several reviewing comments for the authors, please see below.

Line 122: Please write the names of the mentioned species of plants in this sentence in italics and with space between words “Figure 2. Phylogenetic tree of SWEETs from A.thaliana, O. stativa and A.comosus.”

Line 140: Please write the names of the genes in this sentence in italics: “Figure 3. Evolutionary, gene structural and motifs analyses of AcSWEET genes in pineapple.”

Line 143: Please write the names of the genes in this sentence in italics: “Gene structural of AcSWEET genes were identified by TBtools.”

Line 155: Please mark the x-axis on the graphs for 10 genes (AcSWEET1-AcSWEET10) in Figure 4.

Line 172: Why the genes AcSWEET6, AcSWEET8 and AcSWEET17 were not included in the expression analysis?

Lines182-183: Please write the gene and latin names in italics: „Figure 6. Subcellular localization analysis of AcSWEET11 in Nicotiana benthaminana leaves after 2 days of Agrobacterium injection. Scale bars = 20μm.

Line 274: One space between words is extra pleaceDifferent tissues were collected including root, leaf , stem, flower,. I suggest that you delete speace.

Line 279: Please write the names of the mentioned species of plants in this sentence in italics “The protein sequences of SWEET in A. thaliana and O. sativa were downloaded”

Line 302: Please write the word arabidopsis in lower case in this sentence: The amino sequences of SWEET genes in pineapple, Arabidopsis and rice were“.

Lines 325-326: Please italicize the word Аgrobacteium in this sentence: Plasmid 35S::AcSWEET11-EGFP was transformed into tomatoes by Agrobacterium strain…“

Author Response

Line 122: Please write the names of the mentioned species of plants in this sentence in italics and with space between words “Figure 2. Phylogenetic tree of SWEETs from A.thaliana, O. stativa and A.comosus.”

Answer : Thanks for your comments, after carefully check, the names of species plants have changed in italics.

Line 140: Please write the names of the genes in this sentence in italics: “Figure 3. Evolutionary, gene structural and motifs analyses of AcSWEET genes in pineapple.”

Answer : Thanks for your comments, AcSWEET genes have been italicized in Figure 3.

Line 143: Please write the names of the genes in this sentence in italics: “Gene structural of AcSWEET genes were identified by TBtools.”

Answer : Thanks for your comments, the names of AcSWEET have been italicized.

Line 155: Please mark the x-axis on the graphs for 10 genes (AcSWEET1-AcSWEET10) in Figure 4.

Answer : Thanks for your comments, the x-axis were marked in Figure 4.

Line 172: Why the genes AcSWEET6AcSWEET8 and AcSWEET17 were not included in the expression analysis?

Answer : Thanks for your comments, the expression of ‘AcSWEET6’, AcSWEET8 and 

AcSWEET17 have not yet detected in specific tissues. And these results were discussed in discussion section.

Lines182-183: Please write the gene and latin names in italics: „Figure 6. Subcellular localization analysis of AcSWEET11 in Nicotiana benthaminana leaves after 2 days of Agrobacterium

injection. Scale bars = 20μm.“

Answer : Thanks for your comments, the gene and latin names have been italicized.

Line 274: One space between words is extra pleace „Different tissues were collected including root, leaf , stem, flower,“. I suggest that you delete speace.

Answer : Thanks for your comments, the speace of ‘leaf ’, have been deleted.

Line 279: Please write the names of the mentioned species of plants in this sentence in italics “The protein sequences of SWEET in A. thaliana and O. sativa were downloaded”

Answer : Thanks for your comments, the names of species plants have changed in italics.

Line 302: Please write the word arabidopsis in lower case in this sentence: „The amino sequences of SWEET genes in pineapple, Arabidopsis and rice were“.

Answer : Thanks for your comments, ‘Arabidopsis ’ were amended to  ‘arabidopsis ’.

Lines 325-326: Please italicize the word Аgrobacteium in this sentence: „Plasmid 35S::AcSWEET11-EGFP was transformed into tomatoes by Agrobacterium strain…“

Answer : Thanks for your comments, ‘Agrobacterium ’ have been italicized.

Reviewer 2 Report

Title: Genome-wide identification and expression patterns of AcSWEET family in pineapple and AcSWEET11 mediated sugar accumulation

Comments to the author:

They are listed as follows:

The author touched on the AcSWEET gene family identification in pineapple and it is very interesting. This research is very important for proving more suggestions for sugar  accumulation in  pineapple fruit development. However, there are a lot of small mistakes that need to be corrected. They are listed as follows:

1.     Table S2: The primer format should remain the same, which means 5’ and 3’ should all be added or deleted. The ‘primer’ should be ‘primers’.

2.     Illustration of supplementary figures and tables: it shows spelling and punctuation mistakes. Please correct them.

3.     All the species’ names need to be in italics. Please double-check. For example, P4, L122; ‘A.thaliana, O.  stativa and A.comosus’.

4.     P5, L141; ‘candidated’ should be ‘candidate’.

5.     P5, L142; ‘AcSWEET’ or ‘AcSWEETs’ should be uniform. P5, L145; ‘analysed’ should be ‘analyzed’.

6.     P4, L135; ‘motif  5’ should be ‘motif5’.

7.     P6, L158; ‘vertical lines’ is confusing.

8.     P6, Figure 4; the tissue-specific expression results of ‘AcSWEET6’ and ‘AcSWEET17’ are missing, and the error bars are missing, which need to be added.

9.     Figure 6 is difficult to prove the conclusion, and the photo needs to be retaken.

10.  P7, L176; ‘35S:AcSWEET11-EGFP vector’ didn’t mention the whole vector’ s information. It should be a binary vector for infiltrating in Nicotiana benthamiana.

11.  P7, L184; To prove ‘Overexpression of AcSWEET11enhances the accumulation of soluble sugar’, the transgenic pineapple of AcSWEET11 overexpression needs to be constructed and detect the tissue-specific soluble sugar level to confirm your conclusion. The transgenic callus needs to be proved are transgenic, which means the GFP-positive photos should be taken and add in the figures. The western blot results are also required for confirming your transgenic callus.

12.  In Figure 7, the panel A photo is not taken under the same light and the photos between WT and line 15-1 are not showing a big difference. In panel B, the soluble sugar level is not a great difference, which can’t prove your conclusion. Also, the soluble sugar level in the root, stem, leaf, flower, and any other tissues.

In conclusion, the bioinformatic analysis of this research is very impressive. However, the molecular proof is not strong enough. I suggest the author resubmit to the other journals.

Author Response

  1. Table S2: The primer format should remain the same, which means 5’ and 3’ should all be added or deleted. The ‘primer’ should be ‘primers’.

Answer : Thanks for your comments, after carefully check, the primer format should remain the same. The ‘primer’ was mended to ‘primers’.

  1. Illustration of supplementary figures and tables: it shows spelling and punctuation mistakes. Please correct them.

Answer : Thanks for your comments, after carefully check, we have corrected the spells and mistakes.

  1. All the species’ names need to be in italics. Please double-check. For example, P4, L122; ‘A.thaliana, O.  stativa and A.comosus’.

Answer : Thanks for your comments, after carefully check, the names of species plants have changed in italics.

  1. P5, L141; ‘candidated’ should be ‘candidate’.

Answer : Thanks for your comments, ‘candidated’ were amended to ‘candidate’.

  1. P5, L142; ‘AcSWEET’ or ‘AcSWEETs’ should be uniform. P5, L145; ‘analysed’ should be ‘analyzed’.

Answer : Thanks for your comments, ‘AcSWEET’ or ‘AcSWEETs’ were uniform ‘AcSWEETs’ . And  ‘analysed’ were amended to ‘analyzed’.

  1. P4, L135; ‘motif  5’ should be ‘motif5’.

Answer : Thanks for your comments, ‘motif  5’ was amended to ‘motif5’.

  1. P6, L158; ‘vertical lines’ is confusing.

Answer : Thanks for your comments, ‘vertical lines’ represented as mean of SE and we were amended to ‘standard error bars’.

  1. P6, Figure 4; the tissue-specific expression results of ‘AcSWEET6’ and ‘AcSWEET17’ are missing, and the error bars are missing, which need to be added.

Answer : Thanks for your comments, the expression of ‘AcSWEET6’ and ‘AcSWEET17’ have not yet detected in specific tissues. And these results were discussed in discussion section.

  1. Figure 6 is difficult to prove the conclusion, and the photo needs to be retaken.

Answer : Thanks for your comments, we have clarified the conclusion, and replaced the photo.

  1. P7, L176; ‘35S:AcSWEET11-EGFP vector’ didn’t mention the whole vector’ s information. It should be a binary vector for infiltrating in Nicotiana benthamiana.

Answer : Thanks for your comments, we have supplied the information of 35S:AcSWEET11

-EGFP vector, and the whole vector's information was described in the part of 4.6.

  1. P7, L184; To prove ‘Overexpression of AcSWEET11enhances the accumulation of soluble sugar’, the transgenic pineapple of AcSWEET11 overexpression needs to be constructed and detect the tissue-specific soluble sugar level to confirm your conclusion. The transgenic callus needs to be proved are transgenic, which means the GFP-positive photos should be taken and add in the figures. The western blot results are also required for confirming your transgenic callus.

Answer : Since transgenic of pineapple need take a long time,  transient transformation of pineapple callus were used to confirm preliminary function of AcSWEET11 in this study. Also, the transgenic of pineapple are currently underway. Moreover, the GFP-positive photos of transient callus have taken and add in the figure S3. Since the transient calluses were obtained from transient transformation not transgenic callus, the western blot assay were not performed.

  1. In Figure 7, the panel A photo is not taken under the same light and the photos between WT and line 15-1 are not showing a big difference. In panel B, the soluble sugar level is not a great difference, which can’t prove your conclusion. Also, the soluble sugar level in the root, stem, leaf, flower, and any other tissues.

Answer : The photo of figure 7A was replaced. And the statistical analysis of soluble sugar contents were carried out, showing a difference between WT and line 15-1. Besides, we are concerned about the function of AcSWEET11 on fruit development, therefore, the sugar contents were not detected in other tissues.

In conclusion, the bioinformatic analysis of this research is very impressive. However, the molecular proof is not strong enough. I suggest the author resubmit to the other journals.

Reviewer 3 Report

I have gone through the manuscript entitled “Genome-wide identification and expression patterns of AcSWEET family in pineapple and AcSWEET11 mediated sugar accumulation”. In this manuscript the authors have identified 17 AcSWEETs genes distributed across the the whole genome in pineapple. These were validated in different tissues for their expression analysis. Among the identified AcSWEETs genes , AcSWEET11 gene was validated to be  important for fruit development and accounting for accumulating high sugar content. The authors have also validated the function of this gene in tomato transgenically and found that this gene is responsible for sugar accumulation. The manuscript is over all written well. However I suggest the authors to add conclusion.

Line no. 187 detected should be replace with detect.

Author Response

Line no. 187 detected should be replace with detect.

Answer : Thanks for your comments, ‘detected’ were amended to ‘detect.

Reviewer 4 Report

This study discussed the expression pattern of the AcSWEET family and their roles in sugar accumulation in pineapple. Although there are some interesting results, some of the following aspects of the manuscript need to be clarified and corrected for further improvement. 

L. 17 What is the difference between SWEETs and AcSWEETS?

L.24 Specify the further study.

L.28-29 Unclear!

L. 36 Plants?

L. 41 Please change "namely" to "including". 

L. 63-64 Please use past tense. 

L. 66 Move the ref after Ko et al.

L. 68-69 Unclear! Clarify it.

L. 70 Please add the common name for the Averrhoa carambola L.

L. 74. Move the ref after Hao et al.

L. 81-83 Rephrase this sentence to show the objective and hypotheses. 

L. 103 AcSWEET5 was located in the chloroplast and nucleus. clarify it. 

L. 104 in the cell

L. 112 You can use the common name for them after you have shown their scientific name in the introduction. Please revise them below. 

L. 130 exons

L. 157 Please add statistical analysis to show the difference in the relative expression level among different compartments.

L. 160 affected

L. 167 their expression 

L. 185 an embryogenic

L. 187 detect

L. 189 a normal callus

L. 197 Please show the difference in the content of soluble sugar in mature fruit between WT and 15-1.

L. 199 Significantly?

L. 205 Where are the values in Fig. 7B? 

L. 212 were

L. 216 Remove the ref after Hao et al.

L. 217 delete "then".

L. 224 the number

L. 225 Which studies?

L. 226-229 Please use the past tense.

L. 232 Unclear!

L. 243 What is analysis?

L. 254 improved

L. 256 mediated

L. 257 Delete "depending on"

L. 260 increased

L. 332 Please provide the statistical analysis.

Author Response

  1. 17 What is the difference between SWEETs and AcSWEETS?

Answer : Thanks for your comments, "AcSWEETs" is specifies in pineapple.

L.24 Specify the further study.

Answer : Thanks for your comments, further study was specified.

L.28-29 Unclear!

Answer : Thanks for your comments, we have clarified the sentence.

  1. 36 Plants?

Answer : Thanks for your comments, "plates" have mended to "plants".

  1. 41 Please change "namely" to "including". 

Answer : Thanks for your comments, "namely" have changed to "including".

  1. 63-64 Please use past tense. 

Answer : Thanks for your comments, the sentence have changed to past tense.

  1. 66 Move the ref after Ko et al.

Answer : Thanks for your comments, the reference have been moved after Ko et al.

  1. 68-69 Unclear! Clarify it.

Answer : Thanks for your comments, we have clarified the sentence.

  1. 70 Please add the common name for the Averrhoa carambola L.

Answer : Thanks for your comments, the common name of Averrhoa carambola L. was added.

  1. 74. Move the ref after Hao et al.

Answer : Thanks for your comments, the reference have been moved after Hao et al.

  1. 81-83 Rephrase this sentence to show the objective and hypotheses. 

Answer : Thanks for your comments, the sentence have been rephrased to show the objective and hypotheses.

  1. 103 AcSWEET5 was located in the chloroplast and nucleus. clarify it. 

Answer : Thanks for your comments, the location of AcSWEET5 have clarified.

  1. 104 in the cell

Answer : Thanks for your comments, "the cell" was mended to "in the cell ".

  1. 112 You can use the common name for them after you have shown their scientific name in the introduction. Please revise them below. 

Answer : Thanks for your comments, the scientific name was mended to common name.

  1. 130 axons

Answer : Thanks for your comments, " exons " was mended to "axons".

  1. 157 Please add statistical analysis to show the difference in the relative expression level among different compartments.

Answer : Thanks for your comments, the relative expression level were carried out a statistical analysis.

  1. 160 affected

Answer : Thanks for your comments, "affect" was mended to "affected".

  1. 167 their expression 

Answer : Thanks for your comments, "its expression" was mended to "their expression".

  1. 185 an embryogenic

Answer : Thanks for your comments, " embryogenic " was mended to "an embryogenic".

  1. 187 detect

Answer : Thanks for your comments, Spelling was checked.

  1. 189 a normal callus

Answer : Thanks for your comments, "normal callus" was mended to " a normal callus ".

  1. 197 Please show the difference in the content of soluble sugar in mature fruit between WT and 15-1.

Answer : Thanks for your comments, the difference in the content of soluble sugar in mature fruit between WT and 15-1 was showed.

  1. 199 Significantly?

Answer : Thanks for your comments, we have carried out the analysis of statistical.

  1. 205 Where are the values in Fig. 7B? 

Answer : Thanks for your comments, the values were added in Fig. 7B.

  1. 212 were

Answer : Thanks for your comments, "are" was mended to " were ".

  1. 216 Remove the ref after Hao et al.

Answer : Thanks for your comments, the reference have been moved after Hao et al.

  1. 217 delete "then".

Answer : Thanks for your comments, "then" was deleted.

  1. 224 the number

Answer : Thanks for your comments, "numbers" was mended to "the numbers".

  1. 225 Which studies?

Answer : Thanks for your comments, we have clarified the sentence.

  1. 226-229 Please use the past tense.

Answer : Thanks for your comments, the sentence have changed to past tense.

  1. 232 Unclear!

Answer : Thanks for your comments, we have clarified the sentence.

  1. 243 What is analysis?

Answer : Thanks for your comments, we have clarified the sentence.

  1. 254 improved

Answer : Thanks for your comments, " improves" was mended to " improved".

  1. 256 mediated

Answer : Thanks for your comments, " mediates" was mended to " mediated".

  1. 257 Delete "depending on"

Answer : Thanks for your comments, "depending on" was deleted.

  1. 260 increased

Answer : Thanks for your comments, " increases" was mended to " increased".

  1. 332 Please provide the statistical analysis.

Answer : Thanks for your comments, the method of statistical analysis was provided.

Round 2

Reviewer 2 Report

1.     At least, the western blotting needs to be done in transgenic tomatoes to prove the  true transgenic events.

2.     Figure S3 are not clear enough to prove the GFP-positive callus. I only can see several red fluorescent spots. And for transient expression, the expression of AcSWEET11 can be detected via western blot and the related results are required.

3.     The final version of the manuscript without remarks are required and there are still a lot of small mistakes in it that needs to be amended.

Author Response

Comments and Suggestions for Authors

  1. At least, the western blotting needs to be done in transgenic tomatoes to prove the true transgenic events.

Answer : Thanks for your comments, because of the restriction of experiment condition, detection of transgenes in tomatoes by PCR analysis for the hygromycin B gene to prove the true transgenic events.

  1. Figure S3 are not clear enough to prove the GFP-positive callus. I only can see several red fluorescent spots. And for transient expression, the expression of AcSWEET11 can be detected via western blot and the related results are required.

Answer : Thanks for your comments, the GFP-positive callus was marked by red arrows. And because of the restriction of experiment condition, the expression of AcSWEET11were detected by qPCR in normal calluses and transient calluses.

  1. The final version of the manuscript without remarks are required and there are still a lot of small mistakes in it that needs to be amended.

    Answer : Thanks for your comments, after carefully check, grammar errors were amended.

Reviewer 4 Report

Overall, the authors addressed most of the comments I raised. However, I have minor comments which should be addressed. 

L. 17 in the sugar accumulation?

L. 70 where GmSWEET15...

L. 167 What do asterisks mean? It is unclear to me. You can use letters to show the difference. 

L. 216 Replace the even with significantly.

L. 222 In Fig. 7B, why did you remove the comparison between green vs. mature in 15-1? You should keep it.

L. 272-275 There are some grammar errors in this sentence. Modify it.

Author Response

Comments and Suggestions for Authors

Overall, the authors addressed most of the comments I raised. However, I have minor comments which should be addressed. 

  1. 17 in the sugar accumulation?

Answer : Thanks for your comments, ‘during pineapple fruit development’ were amended to  were amended to ‘in the sugar accumulation’.

  1. 70 where GmSWEET15...

Answer : Thanks for your comments, we have amended to ‘where GmSWEET15’.

  1. 167 What do asterisks mean? It is unclear to me. You can use letters to show the difference. 

Answer : Thanks for your comments, we have used letters to show the difference.

  1. 216 Replace the even with significantly.

Answer : Thanks for your comments, ‘even’ were amended to ‘significantly’.

  1. 222 In Fig. 7B, why did you remove the comparison between green vs. mature in 15-1? You should keep it.

Answer : Thanks for your comments, the comparison between green vs. mature in 15-1 was added.

  1. 272-275 There are some grammar errors in this sentence. Modify it.

Answer : Thanks for your comments, after carefully check, grammar errors were amended.

Round 3

Reviewer 2 Report

1.     P4, L117; “MEG6” is wired. Please correct all the wrong spelling in this manuscript.

2.     Supplementary figure s3 and supplementary figure s4 are not shown in the manuscript. If they are no more needed, please delete them from the supplementary figures.

Author Response

  1. P4, L117; “MEG6” is wired. Please correct all the wrong spelling in this manuscript.

Answer : Thanks for your comments, after carefully check, all the wrong spelling were corrected.

  1. Supplementary figure s3 and supplementary figure s4 are not shown in the manuscript. If they are no more needed, please delete them from the supplementary figures.

Answer : Thanks for your comments, after carefully check, the redundant results were deleted in supplementary figure S3 and the figure S4 were deleted.